# A Bayesian Theory of Conformity in Collective Decision Making

**Koosha Khalvati**
Paul G. Allen School of CSE
University of Washington
koosha@cs.washington.edu

**Saghar Mirbagheri**
Department of Psychology
New York University
sm7369@nyu.edu

**Seongmin A. Park**
Center for Mind and Brain
University of California, Davis
park@isc.cnrs.fr

**Jean-Claude Dreher**
Neuroeconomics Lab
Institut des Sciences Cognitives Marc Jeannerod
dreher@isc.cnrs.fr

**Rajesh P. N. Rao**
Paul G. Allen School of CSE &
Center for Neurotechnology
University of Washington
rao@cs.washington.edu

## Abstract

In collective decision making, members of a group need to coordinate their actions in order to achieve a desirable outcome. When there is no direct communication between group members, one must decide based on inferring others' intentions from their actions. The inference of others' intentions is called "theory of mind" and can involve different levels of reasoning, from a single inference of a hidden variable to considering others partially or fully optimal and reasoning about their actions conditioned on one's own actions (levels of "theory of mind"). In this paper, we present a new Bayesian theory of collective decision making based on a simple yet most commonly observed behavior: conformity. We show that such a Bayesian framework allows one to achieve any level of theory of mind in collective decision making. The viability of our framework is demonstrated on two different experiments, a consensus task with 120 subjects and a volunteer's dilemma task with 29 subjects, each with multiple conditions.

## 1   Introduction

Collective decision making is critical for survival in animals that forage as a group [1]. Even though humans are not "hunter-gatherers" any more, collective decision making has remained a crucial element of modern human society, as exemplified by the practice of trial by jury [2, 3]. Group decision making can become extremely challenging when there is no communication between group members, such as in tasks requiring anonymous consensus or volunteers. In these situations, players have to infer the intentions of others from their actions before making their own decisions.

Conformity or aligning one's actions with other group members is a behavior that has been widely observed in group decision making by biologists and psychologists [4, 5, 6], for example, in developing social norms [7, 8]. In fact, even in competitive situations, humans may mimic their opponent's behavior unintentionally [9]. In a collective decision making task, by definition, at least some amount of cooperation between different group members is required for producing utility. Conformity provides a mechanism for cooperation. However, in many situations, there is also some amount of competition between group members, making them cooperate strategically. For example, different players might prefer different outcomes. In these cases, additional processes, such as prediction of the effect of one's action on others, are required for utility maximization.

The future state of mind of other group members and consequently their actions can be linked to one's own current action if the group utilizes conformity. This link can be used by a player to select actions that produce maximum total utility. The ability to infer others' intentions, known as theory of mind (ToM), has been observed in humans extensively [10]. Humans are believed to be able to infer others' current and even future states of the mind during social interactions [11]. While there exist some studies suggesting the connection of theory of mind and conformity in social decision making, e.g., the opportunity for reciprocity in multi-round games [12, 13], there is no mathematical framework or quantitative analysis demonstrating this connection.

Here we present a Bayesian framework based on conformity and theory of mind for expected reward maximization in collective decision making. First, we present a Bayesian model of conformity as the basis for a framework for collective decision making. Then, we show how a (meta-)Bayesian agent can make better decisions (in terms of total utility gain) in complicated tasks with the presence of competitiveness between group members by reasoning about the belief of other Bayesian agents that utilize conformity. In addition, we show this framework can be extended to model different "levels of theory of mind" in collective decision making. Following the terms used in the literature on two-person interactions [14, 15], a Bayesian agent in our framework that utilizes conformity has level-0 ToM, a (meta-)Bayesian agent that reasons about other Bayesian agents that utilize conformity has level-1 ToM, a (meta-)Bayesian agent that reasons about other (meta-)Bayesian agents that reason about Bayesian agents that utilize conformity has level-2 ToM, and so on. This framework is a generalization of our previous level-1 ToM Bayesian model that explained human behavior in a group decision making task [16] to multiple levels of theory of mind and consequently, a broader range of collective decision making tasks.

We tested our framework on two different collective decision making experiments involving human subjects: a consensus task and a volunteer's dilemma task. Our normative Bayesian framework explained and predicted human behavior well on both of these tasks. Moreover, the level of theory of mind that the subjects utilized in the experiments was aligned with the components of the task and information such as the utility function of others. We also present an analysis of the "level of theory of mind" at higher levels not observed in the experiments and show convergence to a Nash equilibrium in collective decision making [17].

## 2 Problem Definition and the Bayesian framework

We investigate the problem of collective decision making with $N > 2$ players and multiple rounds. In each round, the players choose their actions simultaneously and then, all actions in that round are shown to all players, anonymously. The same set of actions is available to each player. More importantly, the reward (utility) of each player in each round depends only on their own action and how many of each of the available actions was selected by others in that round. For simplicity, we assume that the number of possible actions is 2, i.e., the set of actions $A = \{a1, a2\}$.

### 2.1 Bayesian Conformity: Matching the Group

Conformity is matching the behavior of the whole group, i.e., if the probability of choosing action $a1$ by an average group member is $\theta$, a player using conformity as their strategy should choose $a1$ with probability $\theta$ as well. This probability is not observable to the player and each round only provides indirect information about this latent variable via the actions of all $N$ players. Specifically, if the probability of choosing $a1$ by an average group member is $\theta$, the likelihood of observing $m$ $a1$ actions in total from the other $N - 1$ players in a round is given by the Bernoulli probability density function:

$$P(m|\theta) = \binom{N-1}{m} \theta^m (1-\theta)^{N-1-m}. \tag{1}$$

As $\theta$ is not observable, we assume the player maintains a "belief" about $\theta$, i.e., they maintain a probability distribution over $\theta$. Because $\theta$ represents a binomial distribution, we express the belief of the player about $\theta$ with a Beta distribution, which is determined by two parameters $\alpha$ and $\beta$: $Beta(\alpha, \beta) : P(\theta|\alpha, \beta) \propto \theta^{\alpha-1}(1-\theta)^{\beta-1}$ with the divisive normalizing constant $B(\alpha, \beta) = \int_0^1 \theta^{\alpha-1}(1-\theta)^{\beta-1}d\theta$.

Given the above equations, the probability of observing $m$ $a1$ actions by other players is given by:

$$P(m|\alpha, \beta) = \int_0^1 P(m|\theta) P(\theta|\alpha, \beta) d\theta = \binom{N-1}{m} \frac{B(\alpha + m, \beta + N - 1 - m)}{B(\alpha, \beta)}. \qquad (2)$$

The player can update their belief about $\theta$ after each round of the game using the Bayes rule, after the $N-1$ actions come from others. Also, the player considers their own action in the update as well, as they are a member of the group. The posterior probability of $\theta$ after observing $m$ actions of $a1$ from all $N$ players is $Beta(\alpha + m, \beta + N - m)$ [18] (Beta distribution is the *conjugate prior* for Bernoulli distribution).

As the task has multiple rounds, the player starts with a prior probability of $Beta(\alpha_0, \beta_0)$, updates it after each round, and uses the posterior probability of $\theta$ as the prior of the next round. This is a Hidden Markov Model (HMM) where the player infers the state of mind of the group about the next action by observing the previous actions [18].

Due to possible changes in other group members' strategies, the most recent observations are often more reliable, thus deserving a larger weight in the inference. We model this by using a decay rate $0 \leq \lambda \leq 1$ for the prior. This means that a prior of $Beta(\alpha, \beta)$ with $m$ out of $N$ actions being $a1$ in the current round results in a posterior of $Beta(\lambda\alpha + m, \lambda\beta + N - m)$. For example, $\lambda = 0$ means that only the most recent round determines the posterior and $\lambda = 1$ considers all previous rounds equally important.

At the beginning of each round, the player chooses an action. According to the principle of conformity, $a1$ should be chosen with probability of $\theta$. As the player has a posterior probability over $\theta$ instead of the exact value, they use the expected value of $\theta$, which is $\alpha/(\alpha + \beta)$ [18]. We model this scenario with an HMM (instead of a decision process - see below) as the player does not consider the effect of their own action on others and the reward function.

In summary, a player that utilizes Bayesian conformity has a prior belief of $Beta(\alpha_0, \beta_0)$ over $\theta$ before the start of a multi-round game, with a decay rate $\lambda$. At round $t \geq 0$, the player chooses $a1$ with probability of $\alpha_t/(\alpha_t + \beta_t)$. Then, after observing everyone's actions in that round, if there are $m_t$ $a1$ actions in total, the belief changes to $Beta(\alpha_{t+1}, \beta_{t+1})$ where $\alpha_{t+1} = \lambda\alpha_t + m_t$ and $\beta_{t+1} = \lambda\beta_t + N - m_t$.

## 2.2 Meta-Bayesian Conformity: Influencing the Group

The model in the previous section ignored the fact that the player's actions can potentially influence the actions of the group on average. If the group members also utilize conformity, one might be able to influence their future actions particularly if $\alpha$ and $\beta$ are small, the decay rate is large, or if the values of $\alpha$ and $\beta$ are close to each other (i.e., $\theta$ is around .5). As a result, a player who takes advantage of this knowledge can increase their total expected reward over the course of the game by leading the group to states that are more rewarding in the later rounds of the game. The idea of selecting actions that maximize the total expected reward transforms the model from one based on HMMs (previous section) to a Partially Observable Markov Decision Process (POMDP) [19].

Formally, a POMDP is defined as a tuple $(S, A, T, Z, O, R, \gamma)$ where $S$ is the set of states, $A$ is the set of possible actions, and $T$ is the Markov transition function determining the next state $s$ given the current state $s'$ and action $a$, i.e. $P(s|s', a)$. $Z$ is the set of possible observations and $O$ is the observation function that determines the probability of an observation given a state: $P(z|s)$. The reward function determines the reward (utility) received by the player in the current round. Finally, $0 \leq \gamma \leq 1$ is the discount factor for future rewards. Starting from an initial belief $b_0$ (prior probability) and similar to the Bayesian update in an HMM, the POMDP model updates its belief $b_t$ at round $t$ based on its action and the observation (actions of other players) in that round. The goal of the POMDP is to find a *policy* mapping beliefs to actions to maximize the total (expected) discounted reward, i.e., $\sum_{t=0}^{\infty} \gamma^t r_t$.

Finding the optimal policy for a POMDP (solving the POMDP) is computationally very expensive. While recently developed methods can estimate the solution reasonably well [20], as in our framework the belief can be captured with a few parameters ($\alpha$ and $\beta$), we solve the POMDP by casting it as a Markov Decision Process (MDP) whose state space is the POMDP model's belief state space [21]. We define our new MDP, $(S, A, T, R, \gamma)$ as follows. The state space is the space of $(\alpha, \beta)$

determining the subject's belief after each round of the game. The actions remain the same as before: ($A = \{a1, a2\}$). As in the previous section, the transition function is defined by probability of observing the actions of others (Equation 2), the player's own action, and the decay rate $\lambda$:

$$T : \begin{cases} P((\lambda\alpha + m + 1, \lambda\beta + N - 1 - m)|(\alpha, \beta), a1) = \binom{N-1}{m} \frac{B(\lambda\alpha+m, \lambda\beta+N-1-m)}{B(\lambda\alpha, \lambda\beta)} \\ P((\lambda\alpha + m, \lambda\beta + N - m)|(\alpha, \beta), a2) = \binom{N-1}{m} \frac{B(\lambda\alpha+m, \lambda\beta+N-1-m)}{B(\lambda\alpha, \lambda\beta)} \end{cases} \tag{3}$$

As specified in the problem definition, the reward of each player in each round only depends on their own action $a_t$ and actions by other players in that round. Thus, the reward function when $A = \{a1, a2\}$ is a function of $m_t$, number of $a1$ actions by others, and $a_t$: $r_t = R(m_t, a_t)$. Since observations are not modelled in MDPs, we calculate the reward for our MDP's state (which is the belief state of the original POMDP) and action as:

$$R((\alpha, \beta), a) = \sum_{m=0}^{N-1} \binom{N-1}{m} \frac{B(\alpha + m, \beta + N - 1 - m)}{B(\alpha, \beta)} R(m, a). \tag{4}$$

With the above definition of the components of the MDP, the optimal policy can be found easily using Bellman's equation and dynamic programming [21]. This optimal policy, $\pi_t^*$, is a function from the MDP's state space to the action space for ($0 \le t < H$) where $H$ is the maximum number of possible rounds (the horizon).

## 2.3 Higher Levels of Theory of Mind

A Bayesian agent that utilizes conformity only infers the state of mind of others without assuming others have the same inference capability as well (level-0 ToM). The POMDP described above assumes others infer the state of mind of group members as well (level-1 ToM) to the extent of matching their probability of actions. We extend this reasoning here to achieve higher levels of theory of mind. A level-$k$ ToM agent is a POMDP agent that assumes others have level-($k-1$) ToM. The Interactive-POMDP (I-POMDP) model is a general framework for modeling other POMDP agents with arbitrary transition, observation, and reward functions [22]. If the rules of the task are conveyed to all players, achieving higher levels of ToM becomes more computationally tractable as the agent uses the same transition and observation functions for all members. A significant practical problem, however, is that the reward function of others is not often known. As a result, modeling higher levels of ToM becomes more plausible when the reward function is (at least mostly) similar for all group members, e.g., in the case of monetary rewards.

If the player uses a common reward function $R^o$ for others in the group, the level-$k$ ToM agent ($k > 1$) is modeled as a POMDP with state space $S = (\alpha, \beta, t)$ where $t$ is the round number ($0 \le t < H$). The action space remains the same: $A = \{a1, a2\}$. The transition function becomes deterministic as follows. Let the current state be $(\alpha, \beta, t)$. If $\pi_{k-1,t}^* = a1$ where $\pi_{k-1,t}$ is the policy of the level-($k-1$) POMDP with the reward function $R^o$, the next state is $(\lambda\alpha + N, \lambda\beta, t + 1)$ for action $a1$ and $(\lambda\alpha + N - 1, \lambda\beta + 1, t + 1)$ for action $a2$. Similarly, if $\pi_{k-1,t}^* = a2$, the next state is $(\lambda\alpha + 1, \lambda\beta + N - 1, t + 1)$ for the action of $a1$, and $(\lambda\alpha, \lambda\beta + N, t + 1)$ for $a2$. Along the same lines, the reward function is:

$$R((\alpha, \beta, t), a) = R\left((N-1)\mathbb{I}(\pi_{k-1,t}^* = a1), a\right). \tag{5}$$

where $\mathbb{I}(x)$ is 1 if event $x$ happens and 0 otherwise. Note that the assumed reward function of others, $R^o$, could be different from the reward function $R$ of the player but in practice, if there is one reward function for others, it probably applies to all group members including the player.

If the player does not know the reward function of others, they could estimate it based on the dynamics of the game. When a player uses higher than level-0 ToM, they know their actions could lead the group towards selecting actions that produce more reward for themselves. As a result, when a level-1 or higher ToM player chooses an action, either the immediate reward for that action is higher for them, or due to the state of the game, that action produces more expected reward despite producing less immediate reward. In the latter case, the chosen action would not change if one assumes a higher reward for it. As a result, for level-$k$ ToM ($k > 1$), when the state is $(\alpha, \beta)$, the player can

divide other players into two groups based on their "preference" (immediate reward) for an action and estimate the reward function of each group separately:

$$R((\alpha, \beta, t), a) = R\left(\frac{(N-1)\alpha}{\alpha + \beta}\mathbb{I}(\pi_{k-1,t}^{1*} = a1) + \frac{(N-1)\beta}{\alpha + \beta}\mathbb{I}(\pi_{k-1,t}^{2*} = a1), a\right). \quad (6)$$

Similar to the common $R^o$ reward function case above, we define $\pi_{k-1,t}^{1*}$ and $\pi_{k-1,t}^{2*}$ as policies of level-$(k-1)$ POMDP model and use the reward function $R^{o1}$ with action $a1$ having a higher immediate reward, and $R^{o2}$ with action $a2$ being the more rewarding action.

## 3 Experimental Results

We tested our framework on the human behavioral data from two different collective decision making experiments. The first was a consensus group decision making task [23] where $N = 6$ or $N = 4$ players need to agree on one among several items presented to them within a limited number of rounds. The second experiment was a multiple-round Volunteer's Dilemma task [24] where $N = 5$ players had to each decide whether to contribute to a pot of monetary units for the public good or not. In each round of each game, every group member got extra reward if and only if at least $k$ players had agreed or contributed. In both of these experiments, each subject played a multiple-round game several times. The set of players did not change during each game. We fit models based on conformity with different levels of ToM to the behavior of each subject and compared the accuracy of the different level models in explaining human behavior. Specifically, in our model fitting, we used a set of free parameters that explain all games of each subject across all the different conditions. Thus, different conditions are modelled naturally in our framework without any parameter tuning for each condition. The accuracy of a model was determined by the similarity between the model's predicted action and the actual action of the subject in each round on average. In other words, if the predicted action of a model was $\hat{a}$ and the real action was $a$ in a round, the average error was the average of the binary error $|\hat{a} - a|$ over all rounds of all games for the subject (accuracy = 1 - average error). In the level-0 ToM, similar to classification methods, and to produce comparable results for higher levels of ToM, the selected action was $a1$ when $\alpha/(\alpha + \beta)$ was more than .5, and $a2$ otherwise.

We also computed the overall accuracy (all levels of ToM combined) for our framework in each task for each subject. In addition to fitting accuracy, we calculated Leave-One-Out Cross Validation (LOOCV) accuracy where at each iteration, the left-out data point was one whole game. We compared both fitting and LOOCV accuracy of a reinforcement-based model-free approach to our framework [25, 26]. In this approach, the player chooses the most rewarding action according to rewards in previous rounds: the agent starts the task with an initial value for each action, chooses the action with the maximum value in each round, and updates its value based on the gained reward in that round with a weight called the learning rate [27]. Details of the method are in the supplementary material.

### 3.1 Consensus Decision Making

We analyzed the behavioral data of [23] with 120 subjects performing consensus decision making in groups of $N = 6$ or $N = 4$ members. Each subject played the game 40 times (20 with 5 other players, 20 with 3 other players). Each game contained multiple rounds; the number of rounds depended on when the players reached an agreement. Each game started with the presentation of two options, e.g, a piece of chocolate and an apple. The subjects had to choose one of them in each round. The game ended when all players chose the same option. Options remained the same throughout the whole game. After each round, each player observed the others' selected actions as red dots under each option (individual actions are anonymous). After the 10th round, if consensus had still not been reached, the game ended with a probability of $25\%$ in each subsequent round. If the players reached a unanimous consensus, they all received the chosen option. If the game ended without a consensus, subjects did not receive anything. Before the experiment, subjects' preference or value for each item (between \$0 to \$4) was determined through a Becker-DeGroot-Marschak (BDM) auction [28]. More details of the task can be found in the original article describing this experiment [23].

### 3.1.1   Model Fitting and Predictions

We analyzed the games that lasted more than 1 round for each subject. As each player did not know the values of other group members for each item, we used the actions from the first round as the prior for the rest of the game, i.e., $\alpha_1 = m_0$ and $\beta_1 = N - m_0$ (equivalent to considering no prior knowledge at the beginning) where $m_0$ is the number of players that chose option 1 in the first round. As a result, for the conformity (level-0) model, there was only one free parameter, the decay rate $\lambda$.

For the POMDP model (level-1), for the prior belief, we used the same approach as the one for the level-0 model above. The reward for each option for each player was set to the value of that item that was determined in the auction process before the experiment, plus a constant value as the reward for winning. This winning value was a free parameter but kept the same for all available items for each subject. Moreover, it was constrained to be between \$0 and \$10 to be comparable with the estimated values of the items. Overall, the level-1 model had two free parameters (decay rate and winning value) for each subject.

For the level-2 ToM model, as the subjects did not know the others' values for each item, we used the second approach explained in our framework, i.e., using two reward functions $R^{o1}$ and $R^{o2}$. We assumed that the subject estimated the value of each action for others as \$2.5 for their favourite option ($a1$ for $R^{o1}$, and $a2$ for $R^{o2}$) and \$1.5 for the other option, plus the same winning value for all players (similar to the level-1 model). The prior was also computed in the same manner as previous levels. Therefore, the free parameters of the level-2 (and higher) models are the same as the level-1 model. More details of model fitting are presented in the supplementary material. As shown in Figure 1a, the behavior of 83 of the 120 subjects in the experiment were better explained by the level-0 model compared to the level-1 model in terms of higher fitting accuracy. The behavior of 30 out of the remaining 37 subjects were better explained by the level-1 model with more than a $5\%$ difference, with level-1 having one more free parameter. Average fitting accuracy was $77\%$ and $72\%$ for level 0 and level-1 models respectively. The level-2 model had a lower fitting accuracy compared to the level-1 model in almost all subjects (Figure 1b). Therefore, we used only the first two levels of ToM for further analysis. A model with a mix of level-0 and level-1, i.e., choosing the more accurate level for each subject, resulted in $80\%$ accuracy ($SD = 0.07$) (Figure 1c). The average LOOCV accuracy of this mixed model was $75\%$ ($SD = 0.12$).

We also fit the model-free reinforcement learning model with three free parameters to the subjects' behavior. One of these free parameters was the winning value described above which was added to the value estimated for each item, the same as our ToM models above. For the learning rate, we used the function $1/(\kappa_0 + \kappa_1 t)$ with $\kappa_0$ and $\kappa_1$ as the free parameters to model the dependency of the learning rate on the round number $t$. The average fitting accuracy of this model was $62\%$ ($SD = 0.13$), significantly worse than the fitting accuracy of both level-0 and level-1 ToM models in our framework (two-tailed paired t-test, compared to level-0: $t(119) = 10.64$, $p < 0.001$, compared to level-1: $t(119) = 8.08$, $p < 0.001$), and thus also significantly worse than the mix of these two ToM models (Figure 1d). In addition, the average LOOCV accuracy of the model-free reinforcement learning model was $49\%$ ($SD = 0.12$), significantly worse than LOOCV accuracy of both the level-0 and level-1 ToM models (two-tailed paired t-test, compared to level-0: $t(119) = 15.57$, $p < 0.001$, compared to level-1: $t(119) = 5.98$, $p < 0.001$), and the mix of the two models (two-tailed paired t-test: $t(119) = 15.71$, $p < 0.001$).

When fit to a subject's behavior only, our framework can predict whether or not the subject thinks the game will end in the next round using Equation 2 in each state. This prediction cannot be fully validated as we do not have access to the subject's real belief. However, we compared the model's prediction to outcomes in the actual games. Our model based on a mix of level-0 and level-1 ToM predicted whether or not the game will end in each round with $76\%$ average accuracy.

### 3.2   Volunteer's Dilemma

We analyzed the behavioral data from a Volunteer's Dilemma task [24] where 29 subjects played 12 games of a multi-round thresholded Public Goods Game (PGG). Each game involved the subject playing 15 rounds within the same group of $N$ players ($N = 5$). At the beginning of each round, 1 monetary unit (MU) was given to each player. Each player could choose between two options: Giving up the 1 MU (contribute), or keeping it (free-ride). If at least $k$ players contributed ("volunteered"), all players got another 2 MUs (public good) in that round. Otherwise, no reward were given. As a

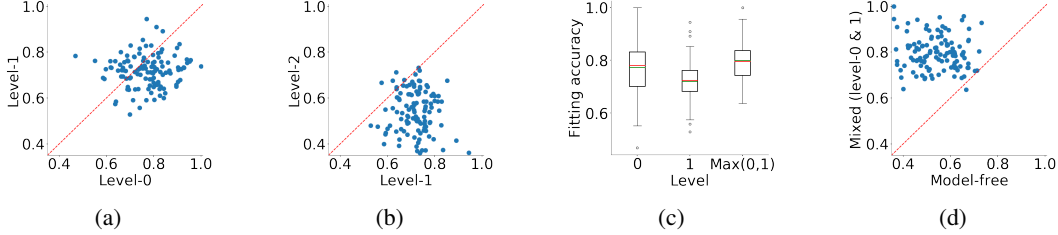

Figure 1: **Different levels of ToM in consensus decision making**. (a) Comparison of level-0 and level-1 ToM fitting accuracy for each subject. The level 0 model explained a greater proportion of the subjects' behavior (see text for details). (b) Comparison of level-1 and level-2 ToM fitting accuracy for each subject. Level-1 explained the subjects' behavior better for almost all of the subjects. (c) Mixing the first two levels of ToM (picking the level with higher accuracy for each subject) increased the accuracy of explaining the subjects' behavior. The green line shows the mean and the red line represents the median in each box plot. (d) Our framework outperformed the reinforcement-based model-free approach for almost all subjects.

result, If the public good was produced (the round was a success), the contributors ended up with 2 MUs in that round while the free-riders had 3 MUs. Similarly, in the case of failure in producing public good, contributors ended up 0 as they gave up their 1 MU while free-riders had 1 MU.

Similar to the consensus experiment, players observed everyone's actions anonymously at the end of each round. The required number of volunteers $k$, was chosen randomly between $k = 2$ and $k = 4$ with equal probability at the beginning of each game. This number was conveyed to the subjects and remained the same through out the whole game. While the subjects thought they were playing with other human players, in contrast to consensus task, they were playing with an algorithm (based on previous human data) that generated the actions of other players. More details can be found in [24].

### 3.2.1 Model Fitting and Predictions

In this experiment, the reward function for all players is the same because monetary reward was used (rather than items of different desirability). As a result, players might use a prior based on their previous experience in life or fictional play, even before the start of the game. For the models for all levels of ToM, there are 3 free parameters in total, i.e., $\lambda$, $\alpha_0$ and $\beta_0$. As seen in Figures 2a and 2b, the level-1 model's fitting accuracy was higher than the level-0 and level-2 models. There is also a strong correlation between accuracy of different levels due to the fact that games with less changes, i.e., consistent contributions or consistent free rides by the subject, make the fit better for all methods.

Due to the higher accuracy of the level-1 ToM model, we compared the model-free reinforcement learning (RL) model only to the level-1 model. The RL model had 5 parameters in total. The first parameter was a reward for generating public good, which was added to the monetary reward. The next two parameters determined the chance of producing public good for $k = 2$ versus $k = 4$, and were used to define the initial Q-value of each action. The final two free parameters determined the learning rate, similar to the consensus task (more details in the supplementary material). The average fitting accuracy of level-1 POMDP model was $84\%$ ($SD = 0.06$) while the average fitting accuracy for the RL model was $79\%$ ($SD = .07$) which is significantly worse than the level-1 model's fitting (two-tailed paired t-test, $t(28) = -6.75$, $p < 0.001$). Also, the average LOOCV accuracy of level-1 POMDP model was $77\%$ ($SD = 0.08$), significantly higher than average LOOCV accuracy of the RL model which was $73\%$ ($SD = .09$) (two-tailed paired t-test, $t(28) = 2.20$, $p = 0.037$).

The level-1 model after fitting can also predict a subject's belief about the success of the group in generating the public good in each round (similar to predicting the end of the game in the consensus task). The level-1 model predicted the success rate of each round with $73\%$ accuracy on average. Moreover, the pattern of this prediction was similar to the actual data when the games for different conditions ($k$s) were compared for each subject (Figure 2c).

Higher levels of ToM in our framework assume deeper levels of optimality in terms of a player's own reward maximization. This optimality, also known as rationality in game theory [29], leads to a Nash equilibrium when the depth increases to infinity [22, 17]. We tested this for our Volunteer's Dilemma task, in which all free-rides is a Nash equilibrium. Specifically, we fit different levels of ToM to the

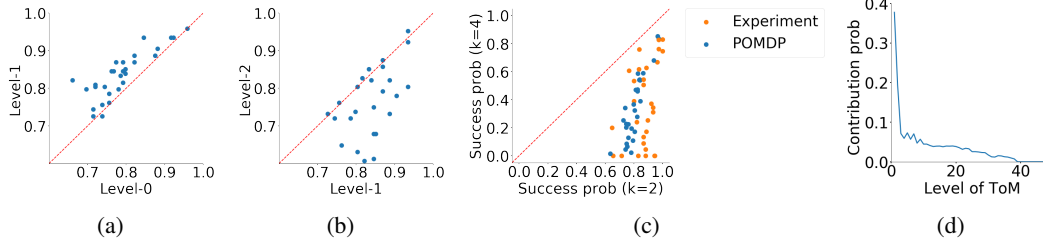

Figure 2: **Different levels of ToM in the volunteer's dilemma task.** (a) Comparison of level-0 model and level-1 ToM model fitting accuracy for each subject. The level-1 model explained the behavior better for almost all of the subjects. (b) Comparison of level-1 and level-2 ToM fitting accuracy for each subject. Level-1 explained the behavior better for almost all of the subjects. (c) The level-1 ToM model could predict the success rate of producing public good in each round quite accurately. The pattern of this prediction (blue circles) matched the reality (orange circles) when the games with different conditions ($k$) were compared to each other for each subject. (d) As the level of ToM becomes higher, the average contribution rate converges to zero, consistent with the fact that a fully rational agent should free ride in all rounds of a volunteer's dilemma game with a known number of rounds (free riding in all rounds is a Nash equilibrium).

data and calculated the average contribution rate of the subject predicted by the model. As seen in Figure 2d, the predicted contribution rate decreased to 0 gradually as the level of ToM increased, despite being fit to a dataset with a contribution rate significantly higher than 0. A player whose only goal is maximizing their own reward does not contribute in the last round as there are no future rounds. In fact, consistent with the principle of conformity, the most important effect of contribution is increasing the contribution rate of others to produce more reward in the future. As the level of ToM increases, the player free-rides in earlier rounds because others (modeled as optimal agents) would be expected to free ride in later rounds. Thus, using higher levels of ToM shifts free-riding towards the first round, decreasing the contribution rate over all rounds to 0 (Figure 2d).

## 4 Discussion

We presented a new Bayesian framework for modeling conformity and theory of mind in collective decision making. To our knowledge, this framework is the first multi-level ToM model for group decision making. Previous models covered only two-person interactions [14, 30, 15, 17, 31] or only a single level of ToM [24, 32]. We demonstrated the viability of our framework using data from two different experiments, each with different conditions. In addition to experimental fits and predictions, we showed that the levels of ToM that explain subject behavior are aligned with the conditions of the two tasks. In the consensus task, the behavior of most of the subjects was explained best by a level-0 ToM model. In this task, different choices could be desirable to different subjects but they knew all players had to pick the same choice in order to finish the current game and gain one of the choices. On the other hand, in the volunteer's dilemma task, while the players might want to produce public good, they knew that having more than the required number of volunteers would be a waste of resources (especially for $k = 2$). Strategic game play and reasoning on current and future intentions of others seems more necessary in the Volunteer's Dilemma task. Consistent with this intuition, nearly all subjects were better fitted with the level-1 ToM model. A level-2 ToM model could not explain the subjects' behavior in our experiments. This was expected due to the computational cost of deep reasoning and lack of observance of higher levels in two-person interaction studies in general [15]. Finally, while we illustrated the approach with only two possible actions, the framework can be easily extendable to more actions simply by using multinomial and Dirichlet distributions [18].

## Acknowledgments

This work was funded by a Templeton World Charity Foundation grant, CRCNS NIMH grant no. 5R01MH112166-03, and NSF grant no. EEC-1028725 and an NSF-ANR Collaborative Research in Computational Neuroscience 'CRCNS SOCIAL POMDP' no.16-NEUC grant. We thank Shinsuke Suzuki and John P. O'Doherty for providing us their data from the consensus task.

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
