[Supplementary Material · k_ToM__NIPS2019___Camera_ready_(1).pdf]

# Supplementary Material for: A Bayesian Theory of Conformity in Collective Decision Making

**Koosha Khalvati**
Paul G. Allen School
University of Washington
koosha@cs.washington.edu

**Saghar Mirbagheri**
Department of Psychology
New York University
sm7369@nyu.edu

**Seongmin A. Park**
Center for Mind and Brain
University of California, Davis
park@isc.cnrs.fr

**Jean-Claude Dreher**
Neuroeconomics Lab
Institut des Sciences Cognitives Marc Jeannerod
dreher@isc.cnrs.fr

**Rajesh P.N. Rao**
Paul G. Allen School
University of Washington
rao@cs.washington.edu

This document contains the details of our algorithms and fitting processes.

## 1 Model-Free Approach: Q-Learning

An alternative approach to reasoning about the intention of others in multi-agent decision making is reward-based learning. In this approach, in each round, the player chooses the most rewarding action based on rewards received in previous trials. Q-learning is one of the most common examples of this approach. In Q-learning, a Q-value is computed for each action, in our case, $Q(a1)$ and $Q(a2)$. Starting from initial values $Q^0(a1)$ and $Q^0(a2)$), in each round, the player chooses the action with the maximum Q-value and updates the Q-value for that action using a weighted average of the current Q-value and the reward received in that round:

$$0 \le t : \begin{cases} \hat{a}^t = argmax_{a \in \{a1,a2\}} Q^t(a) \\ Q^{t+1}(a^t) \leftarrow (1 - \eta^t)Q^t(a^t) + \eta^t r^t \end{cases} \tag{1}$$

where the weight $\eta^t$, called the learning rate, is a function of time (round number).

## 2 Model Fitting: Consensus Decision Making

By using the first round as the prior, the level-0 ToM model had only one free parameter, the decay rate. It was obtained by a grid search with the precision of .25, i.e., $\lambda \in \{0, .25, .5, .75, 1\}$. For level-1 and higher models, we also added a "winning value" $W$, $0 \le W \le 10$, for each subject. If the value of item $k$ was $v_k$ for a subject in the auction, we set the reward for acquiring that item to $W + v_k$ for that subject. Note that $v_k$ is known.

The possibility of ending up in a terminal state without any reward after the tenth round with $25\%$ chance was incorporated into the transition function to match the reality. We used a discount factor of .5 for all subjects, which favors current rewards more than later ones. For level-2 (and potentially higher level) models, we divided the other players into two groups with a value of \$2.5 for the favorite choice and \$1.5 for the other choice (the same $W$ as the player was used for both groups). The reward functions are therefore:

$$\begin{cases} R^{o1}(N - 1, a1) = W + 2.5, & R^{o1}(0, a2) = W + 1.5 \\ R^{o2}(N - 1, a1) = W + 1.5, & R^{o2}(0, a2) = W + 2.5 \end{cases} \tag{2}$$

For the model-free approach, we set the initial value of each choice to $W + v_k$, where $W$ is a free parameter for each subject. Additionally, we used two parameters to model the time-varying learning rate: $\eta^t = \frac{1}{\kappa_0 + \kappa_1 t}$.

## 3   Volunteer's Dilemma

As discussed in the paper, there were 3 free parameters for each subject: $\alpha_0$, $\beta_0$ and the decay rate $\lambda$. As the number of rounds in each game is fixed, we set the discount factor $\gamma$ to 1 for all subjects for level-1 and higher models. In addition, a long horizon of 50 (similar to an infinite horizon) gave a better fit to the behavior of all subjects, consistent with contributing in the last rounds. Also for consistency with the consensus task, we ignored the first round in the fits for all models.

To fit the models at all levels, we used grid search over different values of the parameters. $\alpha_0$ and $\beta_0$ were fit based on a search over integer values from 1 to 200 for each. The decay rate was fit based on a search over real numbers from 0 to 1 with a precision of .1, i.e., $\gamma \in \{0, .01, \ldots, .99, 1\}$. For computational efficiency, we rounded up $\alpha_t$ and $\beta_t$ to integers in all rounds. We used the monetary rewards as the reward function for our ToM models. Adding an extra reward for group success, did not improve the fits in our models.

In the Q-learning model, adding an extra reward for the group success improved the result. Specifically, we made the group reward $G$ a free parameter for each subject in the Q-learning. Moreover, at the beginning of each game, $Q^0(c)$ for the "contribute" action and $Q^0(f)$ for the "free-ride" action were initialized to the expected reward for that action. As this expected reward depends on probability of success, we added two free parameters for this probability, one for each $k$, i.e., $p_2$ and $p_4$. As a result, in a public goods game with $k$ required volunteers, we have:

$$
\begin{cases}
Q^0(c) \leftarrow p_k G \\
Q^0(f) \leftarrow p_k G + 1
\end{cases}
\tag{3}
$$

As in the case of the consensus task, two free parameters were used to describe the learning rate: $\eta^t = \frac{1}{\kappa_0 + \kappa_1 t}$. Importantly, reward after each round $r_t$ also included $G$ for the group reward.

To calculate the average contribution rate at each level, we fitted up to 50-level ToM models to the data. Due to computational limitations, the decay rate was kept 1 for levels higher than 3.