[Reviews · NeurIPS 2019]

Reviewer 1



This paper introduces a Bayesian formulation of Theory of Mind with a focus on conformity. The model is able to represent multi-level reasoning processes in multi round games. The experimental results confirmed that the model provides a better fit to data compared to previous works. The manuscript is very well written and it was a pleasure read. The ideas are well described and motivated (except one section which is mentioned below) and the experiments are well conducted. I have two concerns about the paper. My first concern is about the extent of contributions. It seems to me that the only novel part of the paper is section 2.3 and the rest is very similar to reference [23] cited in the paper. Based on this, it would be important to make this point clear in the paper that the formulation is based on reference [23] and clarify the extend of the contribution in this respect. My other concern is about the implications of the work. The paper shows that the framework can better predict choices of subjects – which is interesting – but the psychological implications of this is not clear in the paper. One potential implication could be inferring the level of reasoning in these kinds of games, but in the last paragraph of the discussion the authors point out that it is not possible to determine the level of reasoning from fit to data. It then would be important to highlight the implications of the model beyond action prediction. Having said this, overall, I think this paper would be of interest to the Neurips audience and I support its acceptance. Some other comments: - In line 154-156 it is a bit unclear why the policy of *others* is assumed to be deterministic (unlike the policy of the agent). Under the non-deterministic policy, the expected next state will be (\lambda \alpha + (N -1)\pi_{k-1,t}(a1) + 1, \lambda \beta (N -1)\pi_{k-1,t}(a2), t+1), which is different from when the policy of others is assumed to be deterministic. It would be good to clarity this part. A similar question is with respect to the reward function defined in equation (5), i.e., instead of using either zero or N-1 for m, the expected number of a1 actions could be used in equation 5. - I was not able to follow lines 162 to 173. It would be great to add further explanations to this part. -In line 69, Bernoulli -> Binomial - Lines 134-137: Please add information about the method used to supplementary materials. - I might have missed this, but please define \pi^* (optimal policy) if it is not defined. ======================================== After author response: Thank you for your response to my comments. My understanding of the author response is that the framework in Ref 23 is indeed similar to the framework in this paper and has used a similar POMDP structure. Given this high similarity, I find it inappropriate to cite this closely related paper (Ref 23) only after page 7 and not to discuss its relations to the current work in terms of the mathematical framework. It should be made a lot clearer that the framework is built upon ref 23 (in sections 2.1, 2,2, intro). From the author response it does not seem that the paper will be revised accordingly in the final version, hence I've changed my score.

Reviewer 2



The authors present a new Bayesian framework to model collective decision processes incorporating both conformity bias and theory of mind processes. They start with a formal model of conformity which they then extend to model the impact a player’s actions may have on the decisions of other players in a collective decision process thereby extending it to allow for inference on the cognitive states of others or theory of mind processes with different depths. The authors also present empirical results of applying their framework as well as a model-free RL model in two different samples and show that there model predicts participants actual choices more accurately. In the following sections, I provide specific feedback with respect to originality, quality, clarity and significance. I hope you will find my comments helpful and constructive. Originality The authors provide an accessible, yet innovative framework to link conformity and theory of mind processes in collective decision making quite elegantly. The model extends past research as it is not restricted to a certain number of players (e.g., two players as in other frameworks) and can be easily scaled to larger action spaces. I believe this work is well written and will have significant impact on understanding collective decision making. Clarity & Quality The paper is very clearly written, well-referenced and still covers both theory and empirical application with respect to an intriguing question. They also include a control model and assess face validity (by assessing, whether models predict the end of the round and the theoretical relationship to the Nash equilibrium in one of the tasks) which I especially appreciated. I only have a few minor questions/suggestions for improvement: - In L. 105 you state that inferring on other players’ mental processes would be especially valuable, when the decay rate is high. My first question is, do you refer to a high decay rate in the sense that only most current results matter or do you refer to a decay rate close to 1, which, as you state in the section above equally implies that all past information are weighted equally. That has not been entirely clear from your description. Also, could you elaborate on why the ‘memory’ of past decisions would change the potential utility of explicitly inferring on other peoples’ mental states, is it because you would know more about other people and thus predict their behavior more accurately knowing the past? - Why did you decide to assess statistical difference between model performances using a t-test. Would a McNemar test not be better suited? - Did you run permutation tests to assess significance of the model performance compared to chance performance (although I believe it is quite clear, it would be good to formally show this as well)? - What is the imbalance of decisions made by the participants, if there is an imbalance, did you take this in your accuracy computation into account (e.g. by computing balanced accuracy), as a high accuracy could result from the prevalence of one particular choice. Significance As I already alluded to, I believe that this work constitutes a theoretical and empirical contribution of great significance. COMMENTS TO THE AUTHORS' RESPONSE: First of all, I wanted to thank you very much for your clarification regarding the significance assessment of the prediction performance. After discussing with my colleagues and reading your response, I still support an acceptance. However, as my colleagues rightfully pointed out, I would strongly urge you to clarify the relationship to ref 23 in the introduction. In your last submission you were giving the false impression that the entire approach is new, which is misleading. Please, reference this work early in your draft and adequately. Despite this hopefully unintentional negligence, I still support acceptance, but I will lower my score, since, while I still find that the paper is very well-written and addresses an important question, this of course reduces the contribution and is, in my view, a failure to aknowledge previous work adequately.

Reviewer 3



Update after author response: I would like to thank the authors for the detailed response which addresses most of my concerns, specifically those relating to the previously unjustified assumptions. In light of that, I am updating my score from 6 to 7. ------- In this paper, the authors propose a probabilistic model of human decision making in a collective scenario. The main proposal considers a simple binary decision-making task, and is based on updating the beta prior of each individual depending on the binomial observation likelihood of collective outcome. The modeling of the task as a POMDP follows naturally by considering the belief states. The authors also talk about higher level theory of mind models, and show that the results from the two experiments are better explained by the proposed model as compared to model-free reinforcement learning. The paper considers a very interesting problem of collective decision making that has bearings on multi-agent models, cognitive science, game theory, etc. The initial development is straightforward and the results seem encouraging. However I had a few concerns and clarification questions about the model: 1. Equations 1 and 2 are only valid if all the other agents decisions are IID (Independent and Identically Distributed) with success-rate parameter \theta. This seems like a far fetched assumption since later we acknowledge that there are individual differences between the agents. How can we justify this IID assumption? 2. Another unsubstantiated assumption (related to #1) is that of every agent starting with the exact same prior. People will come into the task with different experiences and expectations and can’t be assumed to share those parameters. 3. The introduction of the decay rate \lambda is quite ad hoc. Maybe a similar effect can be induced in a more principled way by considering that the other agents adhere to the inferred strategy with probability p and revert back to the prior with probability (1-p). That way events in the past get discounted because it’s more likely that a catastrophic forgetting/reset happened. Is there a mathematical justification for discounting the pseudo counts in the beta distribution otherwise? 4. I was left very confused by the hierarchical model in section 2.3. For example, does line 155 assume all the other (level k-1) agents make the same decisions (a1 or a2)? I didn’t understand the decision making policies of the agents with different levels of theory of mind. It would have been really useful if the authors spent more time explaining this contribution, maybe saving space by being more concise when explaining the well understood beta-binomial updates in 2.1. 5. Line 218 is again a little unprincipled. If the priors are supposed to be beta, then we should adhere to that. A simple change would be assuming Beta (1,1) which is uniform and that will get "overwritten" by the observed data in a few iterations. 6. I am not convinced how easily the results can be reproduced since some of the details seem to have been left unstated. For example, how were the parameter optimized? Jointly using some Bayesian technique or grid-search? If latter, was it joint or sequentially for one parameter at a time? (Edit: I see that some of these are answered in the supplementary material. The authors should add that reference in the main text and can safely ignore my comment.) Overall, I like the central idea of the paper but the proposal seems weakly motivated. There are several assumptions that I found unreasonable but that are not discussed. The central contribution of the paper is confusing to me and there is good scope for improving the presentation. All that said, the results, both experimental and model-fitting, are interesting and intuitive. Furthermore, overlooking the assumptions, the development of theory is reasonable. Based on these observations, I am mildly inclined for the paper to be accepted while acknowledging that there were parts of the manuscript that are still opaque to me.

[Author Response · NeurIPS 2019]

We would like to thank all the reviewers for their insightful comments. Their feedback has helped improve the paper significantly. Changes mentioned in our responses below have been incorporated in the revised version of the paper.

**Reviewer 1:** Regarding the contribution of the paper, our Level-1 theory of mind (section 2.2) was similar to Ref [23] except for the existence of the decay rate and having one set of parameters per subject for all conditions in the game, instead of one set per condition for each subject. However, our framework extends that work by serving as a basis for explaining the rationale behind human contribution by connecting it to conformity (section 2.1) as well as higher levels of theory of mind (section 2.3). The POMDP in Ref [23] can only explain the reward maximization aspect of human behaviour, while our general framework explains why human behaviour is optimal with respect to prosocial evolutionary behaviour and theory of mind. To the best of our knowledge, our paper provides the first formal definition of conformity, as well as higher levels of ToM for large groups. With regard to psychological interpretations, as higher levels of ToM are more complex, even with the same number of free parameters, better fit of a higher ToM does not guarantee its superiority over lower levels. That is not true for the opposite case. Better fit of lower ToM does mean superiority over higher levels. Therefore, while we might not be able to determine the exact level of ToM, we can suggest an upper bound for it. Also, normative models such as ours can be used in the field of computational psychiatry (for example see [1]). Specifically, the difference between ToM level, the prior, or the decay rate in patients and the control group is meaningful. Regarding the deterministic/nondeterministic policy of others and the agent itself, the POMDP model always generates a deterministic policy. In psychology/neuroscience experiments with reinforcement learning/Markov process models, stochasticity is added to the model's generated action by feeding the policy to a probabilistic function (e.g., see [22]). Similar to other classification models, this additional uncertainty does not change the prediction of each action (also mentioned in lines 190-192). It only changes the likelihood function of the model. Therefore, we don't need any new parameters to measure the accuracy of our model. However, if we want to make others' policy nondeterministic (according to the agent), we have to add at least one new free parameter. We will expand this part (especially the last paragraphs) in the final version of the paper.

**Reviewer 2:** Regarding the decay rate, higher decay rate (closer to 1) makes the previous observations and the prior more important. As the reviewer mentioned, this means that others' intention and consequently behaviour is more predictable and influenced by past events. Regarding the statistical tests, there is a good chance that rounds of each game, or even rounds of different games of the same subject are not independent from each other. As a result, to ensure the independence of samples in our statistical test, we used the average accuracy of each subject as one data point. We used the t-test because average accuracy is a continuous value and could be well approximated by a Gaussian distribution for all of our methods. We also ran the McNemar test, taking each round of each game of subjects as one data point. The results were in favor of our conclusions even more than the t-test (lower p-values) probably due to assuming a higher number of independent samples. Also, we compared our method to chance, with the permutation test. For both experiments the p-value was less than 0.001. With regard to choice imbalance, choices are balanced in the consensus task (explained in detail in the original study [22]). In the VD, the number of free-rides was slightly higher (56%). The balanced accuracy of our framework is 83% significantly higher than the model-free method with 75% balanced accuracy ($p < .001$). This means that our framework takes lesser advantage of the bias in the data than the model-free method.

**Reviewer 3**: 1-The reviewer is correct. We should have (and will in the final version) emphasized that this is a reasonable assumption only when others are not tractable due to the anonymity of actions (as in our experiments) or a large number of group members (as most of the real situations such as a jury). In that case, the subject assumes "an average group member" that generates actions because they cannot track individuals. 2- Each subject has their own set of parameters (including prior) in our framework. However, we assume that they "think" others have the same model as themselves. This simplifying assumption of 'you are essentially like me' is justifiable due to computational efficiency and anonymity of players. Moreover, this "false consensus" has been observed experimentally in humans (e.g. see [2]) 3- The concept of a decay rate is equivalent to giving a higher weight to more recent observations (samples). We used the decay rate instead of assigning a larger-than-one weight ($w \geq 1$) to the most recent observation for two reasons. First, to make our fitting methods computationally less expensive. Second and more importantly, we wanted to make our framework more aligned with the concept of decay rate or "leak" in psychology and neuroscience, which is used in decision making studies (e.g., see [3]). 4- We are sorry for the lack of clarity in this part. We will expand it in the final version of the paper. By assuming the same reward function for all players, the subject assumes that all other $N - 1$ players choose the same action. Specifically, in round $t$, if the state (belief of the k-ToM agent) is $(\alpha_t, \beta_t)$, the agent assumes that all other agents choose $\pi^*_{k-1,t}(\alpha_t, \beta_t)$. 5- We totally agree with the reviewer that making the prior $Beta(1, 1)$ is more consistent with the general definition of the prior in the framework. Our current choice, however, is more consistent with the interpretation of $\alpha_t$ and $\beta_t$ as previously experienced samples.

[1]  P. Schwartenbeck and K. Friston. Computational phenotyping in psychiatry: a worked example. *eneuro*, 2016.

[2]  M. Devaine and J. Daunizeau. Learning about and from others' prudence, impatience or laziness: The computational bases of attitude alignment. *PLoS computational biology*, 2017.

[3]  M. Usher and J. McClelland. The time course of perceptual choice: the leaky, competing accumulator model. *Psychological review*, 2001.


[Meta-Review · NeurIPS 2019]

The reviewers and I all think the topic of this paper, conformity in decision making, is of great importance and that the paper is well written. The primary reservation of the reviewers is that the paper overstates the novelty of the modeling contribution -- that the proposed model is almost the same as in ref 23. I think the paper would still make a useful addition to neurips but the authors must moderate and clarify their claims on this point, preferably clearly stating the relationship in the introduction.